# Engineering High-Yield Biopolymer Secretion Creates an Extracellular Protein Matrix for Living Materials

Maria Teresa Orozco-Hidalgo,[a] Marimikel Charrier,[a] Nicholas Tjahjono,[a] Robert F. Tesoriero, Jr.,[c] Dong Li,[a] Sara Molinari,[c] Kathleen R. Ryan,[b,d] Paul D. Ashby,[a] Behzad Rad,[a] Caroline M. Ajo-Franklin[a,c]

[a]Molecular Foundry, Lawrence Berkeley National Laboratory, Berkeley, California, USA

[b]Environmental Genomics and Systems Biology Division, Lawrence Berkeley National Laboratory, Berkeley, California, USA

[c]Department of Biosciences, Rice University, Houston, Texas, USA

[d]Department of Plant and Microbial Biology, University of California Berkeley, Berkeley, California, USA

Maria Teresa Orozco-Hidalgo and Marimikel Charrier contributed equally. To decide authorship order, we used a multicriterion decision-making approach which ranked the importance and relative contribution of each author.

**ABSTRACT** The bacterial extracellular matrix forms autonomously, giving rise to complex material properties and multicellular behaviors. Synthetic matrix analogues can replicate these functions but require exogenously added material or have limited programmability. Here, we design a two-strain bacterial system that self-synthesizes and structures a synthetic extracellular matrix of proteins. We engineered *Caulobacter crescentus* to secrete an extracellular matrix protein composed of an elastin-like polypeptide (ELP) hydrogel fused to supercharged SpyCatcher [SC$^{(-)}$]. This biopolymer was secreted at levels of 60 mg/liter, an unprecedented level of biomaterial secretion by a native type I secretion apparatus. The ELP domain was swapped with either a crosslinkable variant of ELP or a resilin-like polypeptide, demonstrating this system is flexible. The SC$^{(-)}$-ELP matrix protein bound specifically and covalently to the cell surface of a *C. crescentus* strain that displays a high-density array of SpyTag (ST) peptides via its engineered surface layer. Our work develops protein design guidelines for type I secretion in *C. crescentus* and demonstrates the autonomous secretion and assembly of programmable extracellular protein matrices, offering a path forward toward the formation of cohesive engineered living materials.

**IMPORTANCE** Engineered living materials (ELM) aim to mimic characteristics of natural occurring systems, bringing the benefits of self-healing, synthesis, autonomous assembly, and responsiveness to traditional materials. Previous research has shown the potential of replicating the bacterial extracellular matrix (ECM) to mimic biofilms. However, these efforts require energy-intensive processing or have limited tunability. We propose a bacterially synthesized system that manipulates the protein content of the ECM, allowing for programmable interactions and autonomous material formation. To achieve this, we engineered a two-strain system to secrete a synthetic extracellular protein matrix (sEPM). This work is a step toward understanding the necessary parameters to engineering living cells to autonomously construct ELMs.

**KEYWORDS** *Caulobacter crescentus*, engineered living material, extracellular matrix, protein hydrogel, protein secretion, surface structures, surface layer protein, type I secretion

Address correspondence to Marimikel Charrier, mmcharrier@colorado.edu, or Caroline M. Ajo-Franklin, cajo-franklin@rice.edu.

Re-programming the homes that bacteria make for themselves - so we can make living materials

Bacterial cells mold their environment through their extracellular matrix (ECM): a heterogeneous matrix of predominately polysaccharides with a mix of proteins, nucleic acids, and minerals (1). The autonomously produced ECM is dynamic, and bacteria vary its charge, hydrophobicity, porosity, or other properties to assist the cell with

survival in various environments. Biofilm matrices function to facilitate mechanical stress tolerance (2), nutrient sorption, and both genetic and chemical communication (3–5). By interacting with the environment and controlling mass transfer, the matrix affects morphology, resilience, and interspecies interactions (3, 6) of the bacterial community, increasing its overall plasticity.

Engineered living materials (ELMs) attempt to mimic aspects of natural systems, including biofilms, and are poised to dramatically impact the fields of soft matter assembly and structural materials by adding abilities such as self-healing, material synthesis, autonomous assembly, and responsiveness (7). Current synthetic biology tools (8), such as pioneering work with curli fibers (9–11) and bacterial cellulose (12, 13), modulate the endogenous ECM content but are limited in the sequence tunability of the biopolymer in the matrix and do not directly encapsulate individual bacteria, as an extracellular polysaccharide (EPS) layer does. Direct cell encapsulation is largely approached using exogenous addition of polymers and attaching them through adhesive motifs or entrapment (14, 15). These approaches lack the autonomous formation of natural biofilms and thus require energy-intensive processing (15, 16) and added expense. Thus, there is an unmet need for a self-forming yet programmable bacterial ECM.

Limited effort has been made to engineer the EPS layer, mostly because the confounding multistep syntheses of nonlinear polysaccharides (17) make them difficult to program genetically. A more tractable approach to engineering the supramolecular structure of the ECM is to manipulate its protein content. We hypothesize that this simplification of the ECM to a synthetic extracellular protein matrix (sEPM) would result in more programmable interactions, allowing for tunable three-dimensional (3D) structures. Previous research shows that alterations in the composition of polypeptides with hydrogel-like behaviors, such as elastin or resilin, leads to different material properties (18–20). The protein structure, degree of cross-linking, and number of weak interactions are all variables impacted by the peptide sequence (21, 22). In addition, proteinprotein interactions can drive highly specific, selective, and even covalent binding, for example, through the SpyCatcher-SpyTag system (23, 24).

The freshwater bacterium *Caulobacter crescentus* is emerging as a platform for synthetic biology and ELMs (25–29). This bacterium provides multiple advantages as a chassis: it is genetically tractable, is well characterized due to its intriguing dimorphic life cycle (30), strongly adheres to surfaces via its holdfast matrix (31), and has a modifiable proteinaceous surface layer (S-layer) (25, 32). In addition, it is oligotrophic and can flourish with minimal nutrients and in cold temperatures (33). We previously reported the construction of a set of *C. crescentus* variants (25), in which we engineered the S-layer protein, RsaA (34, 35), to display SpyTag, which is one part of the split-enzyme SpyTag-SpyCatcher system (36). These strains covalently ligate SpyCatcher-displaying inorganic nanocrystals, proteins, and biopolymers to the extracellular array at high density (25). With these advantages, *C. crescentus* is well positioned as a chassis for developing a sEPM.

To make the formation of a sEPM autonomous, high-level protein secretion is required. However, secretion of heterologous biopolymers with known material properties has proven challenging for Gram-negative bacteria due to their high aggregation tendency and repetitive sequences (37). While typically type I secretion systems (T1SSs) are considered to have low titers (38), the T1SS in *C. crescentus* has the potential to secrete high heterologous protein titers. The T1SS is endogenously tasked with transporting 10% to 12% of the total cell protein to form the RsaA surface layer (39), and secretion of heterologous enzymes has been demonstrated (40). The T1SS is a one-step transport system that consists of an ABC transporter, membrane fusion protein, and outer membrane protein. The hallmark of T1SS substrates is the necessary C-terminal secretion signal. In addition, they typically include RTX domains with the nine-residue consensus sequence GGxGxDxUx, wherein U is a hydrophobic residue, and these domains are usually involved in $Ca^{2+}$ binding. As calcium is strictly regulated

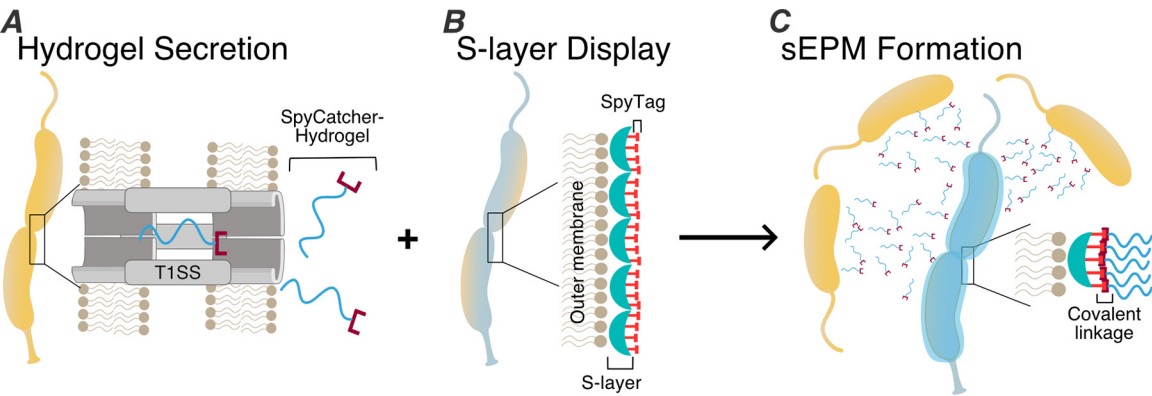

**FIG 1** Design of two-strain consortium that forms a synthetic extracellular protein matrix (sEPM). Schematic of *C. crescentus* secretor strain capable of exporting hydrogel proteins containing SpyCatcher covalent binding motif via a type I secretion system (A), *C. crescentus* displayer strain capable of binding SpyCatcher-hydrogel proteins to engineered RsaA-SpyTag S-layer at high density (B), and encapsulation of displayer cells in the secreted hydrogel protein forming a sEPM (C).

intracellularly at a level lower than required for these proteins to fold, it is presumed that T1SS substrates remain largely unfolded until fully secreted (41). In *C. crescentus*, the RsaD-F$_{a,b}$ T1SS transports the 1,026-amino-acid RsaA surface layer protein (40). This system is unique in that it contains two homologous outer membrane proteins, RsaF$_a$ and RsaF$_b$ (42). The RsaA substrate protein contains an 82-amino-acid secretion signal and six occurrences of the RTX domain. However, either the C-terminal 242 or 336 amino acids are required for maximal secretion of protein (39).

In this work, we put forth a new concept that employs an engineered two-strain system to create a bacterially produced sEPM that subsequently covalently coats the bacterial cell surface. We develop *C. crescentus* strains that use a T1SS to export elastin-like polypeptides (ELPs) or resilin-like polypeptides (RLPs) fused to supercharged SpyCatcher [SC$^{(-)}$] at levels up to 60 mg/liter, the highest level reported for a Gram-negative T1SS. We then demonstrate the sEPM by binding purified SpyCatcher$^{(-)}$-ELP fusion proteins covalently to our engineered SpyTag-displaying strain. Through our secretion efforts, we confirm design guidelines around folding and the isoelectric point required to maximize biopolymer secretion via *C. crescentus*'s T1SS. Thus, this work furthers the understanding of type I secretion and the value of *C. crescentus* as a secretion platform by demonstrating the self-synthesis and self-organization of a rationally designed and tunable synthetic extracellular protein matrix.

(This article was submitted to an online preprint archive [43].)

## RESULTS

**Design and construction of *Caulobacter crescentus* strains to produce an extracellular matrix protein.** We sought to mimic key properties of naturally occurring biofilm matrices while adding the structural and functional flexibility of synthetic ECMs to create a sEPM. This led to several molecular-level design constraints for our sEPM. First, to avoid the need to exogenously add materials, the matrix should be biologically synthesized and secreted extracellularly. Second, to mimic the structure and function of a natural ECM with the potential for mechanical support, trapped hydration, and transport of nutrients and waste products, the matrix should be capable of forming a hydrogel. Third, to open the possibility for spatial patterning and to add robustness, the synthetic matrix should specifically and covalently bind cells of a different strain.

To meet these constraints, we designed a system consisting of two strains of *C. crescentus*: a "secretor" strain capable of synthesizing and secreting a proteinaceous extracellular network with the ability to form a hydrogel matrix (Fig. 1A) and a "displayer" strain capable of binding this protein on its surface at high density (Fig. 1B). When cocultured, we hypothesized that the displayer strain would be coated in an

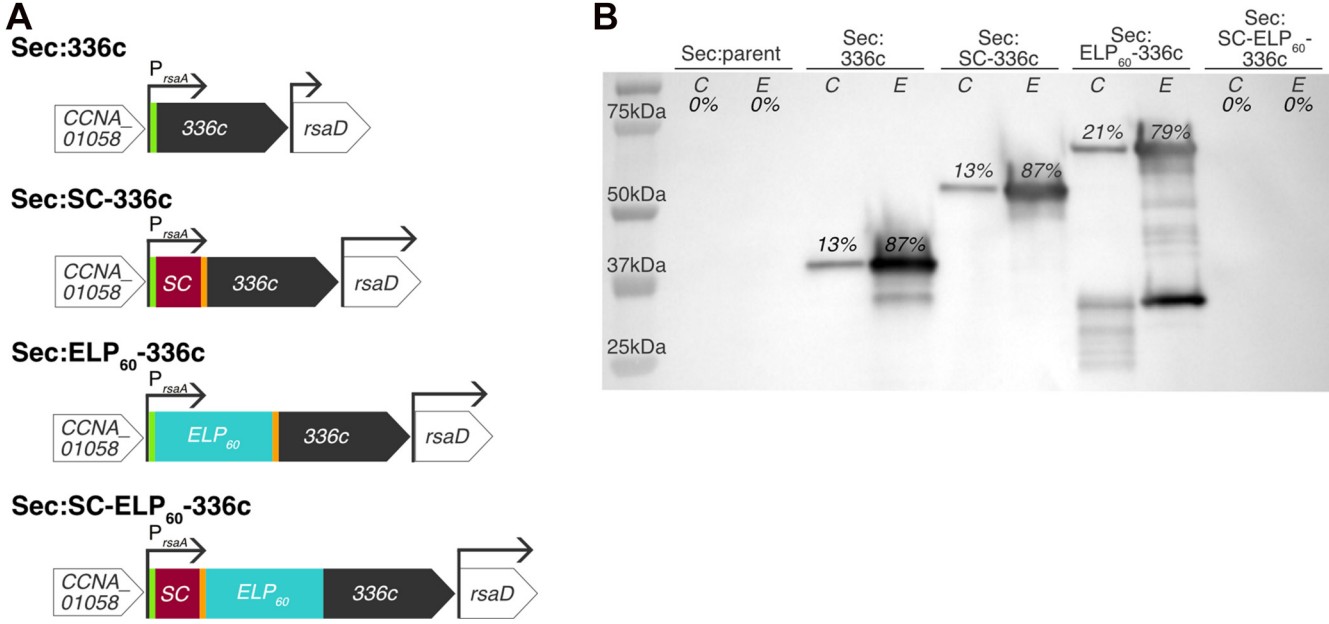

**FIG 2** SpyCatcher-$ELP_{60}$-336c unable to traverse T1SS in *C. crescentus*. (A) Schematic of genome variants for secretion (top to bottom) of the 336c secretion signal, SpyCatcher (SC)-336c control, $ELP_{60}$-336c hydrogel, and full SC-$ELP_{60}$-336c protein. The green segment indicates a FLAG tag and orange segment indicates a streptavidin (Strep) tag. All genes contain the native *rsaA* promoter and 5' untranslated region (UTR). (B) Immunoblot with anti-FLAG antibodies against whole-cell lysate (indicated by C) and extracellular medium (indicated by E) of Sec:parent (first and second lanes, respectively) and Sec:336c (third and fourth lanes), Sec:SC-336c (fifth and sixth lanes), Sec:$ELP_{60}$-336c (seventh and eighth lanes), and Sec:SC-$ELP_{60}$-336c (ninth and tenth lanes). Percentages indicate quantity of internally (0.2% of total fraction) and externally (0.006% of total fraction) detected full-length protein within strains. While full-length proteins are detected in the extracellular medium in the strains expressing 336c, SC-336c, and $ELP_{60}$-336c, the SC-$ELP_{60}$-336c protein is not detected in either the cell pellet or extracellular medium.

autonomously formed sEPM, creating a living material (Fig. 1C). Since we previously engineered versions of displayer strains (25), our major task was to construct secretor strains to secrete an extracellular matrix protein capable of specifically binding to displayer strains.

To create the secretor strain, we designed a heterologous gene containing modules for high-level secretion, antibody-based detection, hydrogel formation, and covalent ligation (Fig. 2A). This heterologous gene includes the native regulatory regions of *rsaA* (*CCNA_01059*), which drive high-level expression, and the C-terminal 336 amino acids of RsaA (notated 336c), which serve as a signal for extracellular secretion (40). While the 336c sequence can be detected via anti-RsaA antibodies (42), we also included an N-terminal FLAG tag so we could probe both termini. Since the displayer strains display SpyTag (44), we utilized its partner, SpyCatcher (SC), as the covalent binding motif. For the hydrogel module, we chose elastin-like polypeptide (ELP), which is a hydrophobic and disordered polymer that readily forms hydrogels (45). This thoroughly characterized recombinant protein has tunable material properties through alteration of the pentapeptide repeat number, inclusion of cross-linking residues, or inclusion of binding or cleavage domains (21, 46, 47). The ELP we utilize is nicknamed $ELP_{60}$ due to its repeat number.

In addition to the heterologous gene that contained all modules, we also created genes with only the 336c signal, without the binding module, and/or without the hydrogel module as controls (Fig. 2A). We introduced these DNA constructs into a modified *C. crescentus* CB15N strain (see Table S1 in the supplemental material). CB15N lacks the holdfast, making it less adherent to surfaces (48), and is a standard strain for studies of *C. crescentus*. In our modified CB15N strain, the native S-layer-associated protein gene (*sapA*) is replaced with a xylose-inducible *mKate2* (49) fluorescent protein gene to eliminate interference with S-layer assembly (50) and to facilitate fluorescent imaging. This parent strain is titled Sec:parent. All genome integrations were confirmed by colony PCR (see Fig. S1B). The resultant secretor strains are titled Sec:SC-hydrogel variant (genotype details available in Table S1).

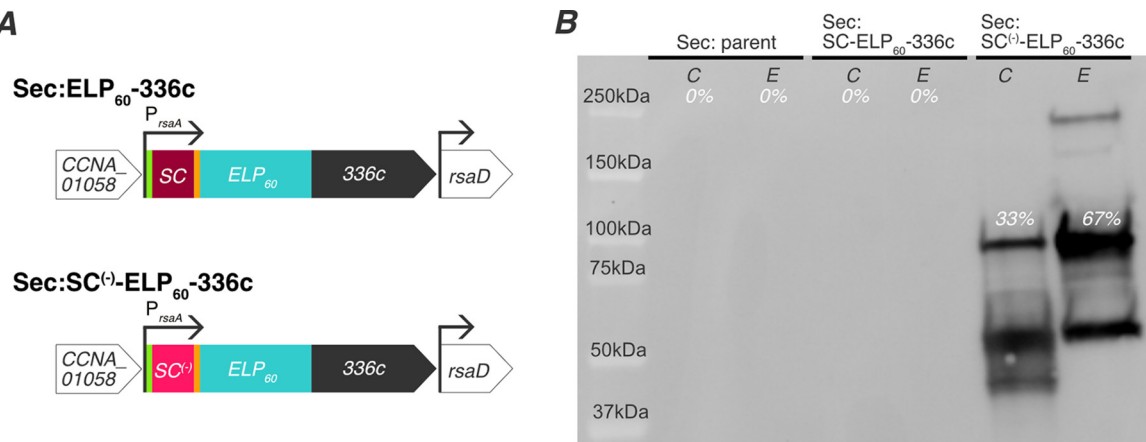

**FIG 3** Supercharged SpyCatcher allows for secretion of synthetic extracellular matrix proteins. (A) Schematic of genome variants for secretion of SpyCatcher (SC)-ELP$_{60}$-336c protein and supercharged SpyCatcher [SC$^{(-)}$]-ELP$_{60}$-336c protein. The green segment indicates a FLAG tag and orange segment indicates a Strep tag. All genes contain the native *rsaA* promoter and 5′ UTR. (B) Immunoblot with anti-FLAG antibodies against the whole-cell lysate (indicated by C) and extracellular medium (indicated by E) of Sec:parent (first and second lanes), Sec:SC-ELP$_{60}$-336c (third and fourth lanes), and Sec:SC$^{(-)}$ELP$_{60}$-336c (fifth and sixth lanes). While the matrix protein is not detected in the Sec:parent and Sec:SC-ELP$_{60}$-336c cultures, the Sec:SC$^{(-)}$-ELP60-336c strain achieves notable levels of secretion.

**Extracellular matrix proteins containing a folded SpyCatcher module are not efficiently secreted.** To test the ability of our engineered strains to secrete synthetic extracellular matrix proteins, we analyzed the proteins present in the cell pellets and extracellular media using immunoblotting with an anti-FLAG antibody (Fig. 2B). We did not detect any bands in the Sec:parent culture (Fig. 2B, first and second lanes), confirming that our assay is specific for FLAG tag-containing proteins. In the Sec:336c culture, we observed a 37-kDa band corresponding to the 336c secretion signal. This band was notably stronger in the extracellular medium than in the cell pellet (Fig. 2B, third and fourth lanes), confirming earlier work that this sequence is sufficient for secretion (40). Similarly, somewhat weaker bands corresponding to SC-336c (57 kDa) and ELP$_{60}$-336c (73 kDa) were observed in extracellular media from the Sec:SC-336c and Sec:ELP$_{60}$-336c cultures, respectively (Fig. 2B, fifth to eighth lanes), indicating that these proteins were secreted. We noted degradation of ELP$_{60}$-336c in both the cell pellet and supernatant fractions, with lower-molecular-weight bands ranging from approximately 53 kDa to 20 kDa. Through densitometry, we confirmed that the majority of expressed protein was present in the extracellular fraction at the time of harvesting (Fig. 2B) (extracellular 336c, 87%; SC-336c, 87%; and ELP$_{60}$-336, 79%). Protein secretion is a dynamic process, and so the measured percentage of secreted protein could vary depending on harvesting times and the relative mass fraction of cells to extracellular media (~1:30). Most critically, we were unable to detect the SC-ELP$_{60}$-336c fusion protein in either the supernatant or cell pellet (Fig. 2, ninth and tenth lanes), indicating that the complete synthetic extracellular matrix protein is either not secreted and subsequently degraded intracellularly or not expressed. Since the heterologous gene is present in the genome (Fig. S1B, lane 5), we suggest that the SC-ELP$_{60}$-336c fusion is unable to traverse the type I machinery because of the folded nature of SpyCatcher (51, 52).

**Replacing SpyCatcher with supercharged SpyCatcher enables secretion of extracellular matrix proteins.** To test the hypothesis that the folded nature of SpyCatcher decreases secretion of fusion proteins, we replaced SpyCatcher with a supercharged variant, SpyCatcher$^{(-)}$ [SC$^{(-)}$], in our synthetic extracellular matrix protein. SC$^{(-)}$ has an additional 12 negative charges introduced into the sequence, which keeps it largely disordered until it binds SpyTag (44). We verified genomic incorporation of SC$^{(-)}$ into the Sec:SC-ELP$_{60}$-336c strain by colony PCR (Fig. S1B, lane 3) and notated it as Sec:SC$^{(-)}$-ELP$_{60}$-336c (Fig. 3A).

We analyzed the protein composition of the Sec:parent, Sec:SC-ELP$_{60}$-336c, and Sec:SC$^{(-)}$-ELP$_{60}$-336c cultures by immunoblotting (Fig. 3B). As expected, no FLAG tag-

mSystems®

containing bands were present in either the cell pellet or supernatant fractions of the Sec:parent culture (first and second lane) or the Sec:SC-ELP$_{60}$-336c strains (third and fourth lanes). However, we detected a band at an apparent molecular weight of 93 kDa in the extracellular medium of Sec:SC$^{(-)}$-ELP$_{60}$-336c cultures (fifth and sixth lanes). While this apparent molecular weight is greater than the predicted molecular weight of SC$^{(-)}$-ELP$_{60}$-336c (86.9 kDa), previous work has demonstrated that SpyCatcher$^{(-)}$ migrates as an ~20-kDa protein, although its expected molecular weight is 14.8 kDa (44). Accounting for this difference, we expect SC$^{(-)}$-ELP$_{60}$-336c to run as a 92-kDa protein, in line with the observed band. This band is also detected with anti-RsaA serum, indicating that this band is indeed the full-length SC$^{(-)}$-ELP$_{60}$-336c (see Fig. S2A). Hence, this confirms our hypothesis that replacing a folded fusion protein (SpyCatcher) with a disordered one [SpyCatcher$^{(-)}$] allows for expression and secretion due to the specificity for unfolded substrates by type I secretion machinery.

There were significant bands at a higher molecular weight (~250 kDa) and a lower molecular weight (~52 and 41 kDa) present in the cell pellet and extracellular medium of Sec:SC$^{(-)}$-ELP$_{60}$-336c cultures (Fig. 3B, fifth and sixth lanes). The higher-molecular-weight band became more pronounced as the protein concentration increased (see Fig. S3). As ELP$_{60}$ is highly hydrophobic and the 336c secretion sequence tends to aggregate (40), we attribute the 250-kDa band to aggregation. The presence of molecular weight bands lower than expected suggests that the protein is sensitive to degradation. Nevertheless, we detected most of the protein at the molecular weight expected for SC$^{(-)}$-ELP$_{60}$-336c, indicating this protein is effectively secreted.

**Extracellular matrix proteins with different hydrogel modules can be secreted at high levels.** Next, we sought to test the modularity of the hydrogel component so that the material properties of sEPM could be tailored. We sought polypeptides that would cover a range of potential material properties while offering variety in hydrophobicity and structure (Fig. 4A). We also required that our selected targets had previously been expressed heterologously and could tolerate sequence insertions.

Using these design criteria, we selected three additional hydrogel-forming polypeptides to incorporate into our sEPM. First, we created a variant of the ELP$_{60}$ sequence, ELP$_{60x}$, that includes a series of lysine and glutamine residues for enzymatic cross-linking with transglutaminase, potentially stiffening resultant ELMs by generating new covalent links between individual ELP$_{60x}$ molecules (53). Our second target, resilin-like polypeptide (RLP$_{12}$) (19), is a recombinant version of an elastomeric protein found in insects that has remarkable extensibility and resilience. While less explored than ELP, several variants of RLP have been produced with modular sequences for tunable material properties, including sites for enzymatic cross-linking and inclusion of biologically active domains (20). Similar to ELP$_{60}$, RLP$_{12}$ is disordered. However, RLP$_{12}$ is more hydrophilic than ELP$_{60}$. Finally, we selected suckerin$_{19}$, a protein found in the sucker ring teeth of squid and cuttlefish. This unique material is highly stiff, with a potential elastic modulus in the gigapascal range (54). It differs from the two other targets not only in stiffness but also in that it is a structured protein of beta-sheets interspersed with amorphous regions. These three new heterologous genes were inserted into the genome in place of the first ELP$_{60}$ gene to create the Sec:SC$^{(-)}$-ELP$_{60x}$-336c, Sec:SC$^{(-)}$-RLP$_{12}$-336c, and Sec:SC$^{(-)}$-suckerin$_{19}$-336c strains, and all were confirmed by colony PCR (Fig. S1B).

Again, we used immunoblotting to identify proteins from these engineered secretor strains. The Sec:SC$^{(-)}$-ELP$_{60}$-336c and Sec:SC$^{(-)}$-ELP$_{60x}$-336c cultures displayed bands of 90 kDa in the extracellular media (Fig. 4B, third and fourth lanes and ninth and tenth lanes, respectively), demonstrating that the ELP$_{60}$ sequence can be tuned while maintaining secretion. Additionally, a band of 81 kDa corresponding to SC$^{(-)}$-RLP$_{12}$-336c was detected in the extracellular fraction (Fig. 4B, fifth and sixth lanes), indicating it also can be secreted. Through densitometry, we confirmed the majority of expressed protein was present in the extracellular fraction for Sec:SC$^{(-)}$-ELP$_{60}$-336c and Sec:SC$^{(-)}$-ELP$_{60x}$-336c at the time of harvesting (Fig. 4B) (extracellular SC$^{(-)}$-ELP$_{60}$-336c, 59%; and SC$^{(-)}$-ELP$_{60x}$-336c, 58%), whereas Sec:SC$^{(-)}$-RLP$_{12}$-336c showed 6% more expressed

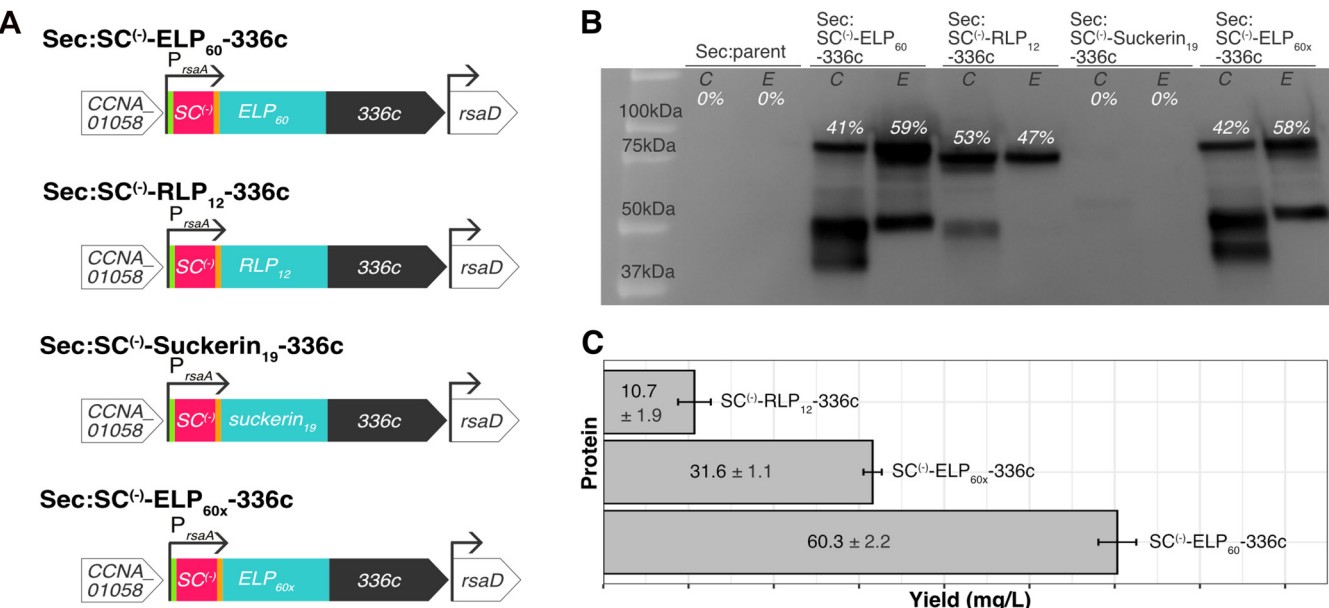

**FIG 4** $SC^{(-)}$-$ELP_{60}$-336c, $SC^{(-)}$-$ELP_{60x}$-336c, and $SC^{(-)}$-$RLP_{12}$-336c are secreted, while $SC^{(-)}$-$suckerin_{19}$-336c is not. (A) Schematic of genome variants for secretion (top to bottom) of SpyCatcher$^{(-)}$-$ELP_{60}$-336c protein, SpyCatcher$^{(-)}$-$RLP_{12}$-336c protein, SpyCatcher$^{(-)}$-$suckerin_{19}$-336c protein, and SpyCatcher$^{(-)}$-$ELP_{60x}$-336c protein. The green segment indicates a FLAG tag and orange segment indicates a Strep tag. All genes contain the native *rsaA* promoter and 5' UTR. (B) Immunoblot with anti-FLAG antibodies of *C. crescentus* whole-cell lysate (indicated by C) and extracellular medium (indicated by E) of Sec:parent (first and second lanes), Sec:$SC^{(-)}$-$ELP_{60}$-336c (third and fourth lanes), Sec:$SC^{(-)}$-$RLP_{12}$-336c (fifth and sixth lanes), Sec:$SC^{(-)}$-$suckerin_{19}$-336c (seventh and eighth lanes), and Sec:$SC^{(-)}$-$ELP_{60x}$-336c (ninth and tenth lanes). Percentages indicate quantity of internally (0.2% of total fraction) and externally (0.006% of total fraction) detected full-length protein within the strain. While full-length proteins are detected in the extracellular medium in the strains expressing $SC^{(-)}$-$ELP_{60}$-336c, $SC^{(-)}$-$RLP_{12}$-336c, and $SC^{(-)}$-$ELP_{60x}$-336c, the $SC^{(-)}$-$suckerin_{19}$-336c protein is only faintly apparent in the cell pellet and not at all in the extracellular medium. (C) Average yields of protein purified from the extracellular media of Sec:$SC^{(-)}$-$ELP_{60}$-336c cultures, Sec:$SC^{(-)}$-$RLP_{12}$-336c cultures, and Sec:$SC^{(-)}$-$ELP_{60x}$-336c cultures. Sec:$SC^{(-)}$-$ELP_{60}$-336c is capable of exporting the highest quantity of a biopolymer by Gram-negative T1SS to our knowledge. Values in the graph represent average protein yields in milligrams per liter of culture ($\pm$ standard deviations [SDs]).

protein in the intracellular fraction (Fig. 4B) (intracellular $SC^{(-)}$-$RLP_{12}$-336c, 53%; extracellular $SC^{(-)}$-$RLP_{12}$-336c, 47%). The expected band for the $SC^{(-)}$-$suckerin_{19}$-336c protein (97.9 kDa) was not detected, but degradation products at lower-molecular-weight bands (53 kDa and 46 kDa) (Fig. 4B, seventh and eighth lanes) were apparent in the cell pellet at very low levels.

It is important to recognize that secretion distribution is also affected by the native protein trafficking dynamics of the cell and the relative mass fraction of the cells and extracellular media. As $SC^{(-)}$-$ELP_{60}$-336c is the sole substrate for the T1SS in our engineered strains (where the native substrate RsaA is knocked out), SDS-PAGE demonstrates that $SC^{(-)}$-$ELP_{60}$-336c is the main protein present in the extracellular fraction at the time of harvesting (Fig. S2B). We confirmed that all of the biopolymer detected in the extracellular fraction was secreted and not released due to cell lysis by immunoblotting with anti-CtrA antibodies to demonstrate that the cell cycle transcriptional regulator (CtrA), an exclusively intracellular protein, only appears in the intracellular fraction (see Fig. S5) (55). Taken together, these results confirm that synthetic extracellular matrix proteins containing different hydrogel polypeptides can be secreted, but that the hydrogel protein sequence has a significant effect on secretion yields.

To determine the total amount (including aggregation and degradation products) of extracellular matrix protein secreted by the engineered strains, we purified the protein from culture media using ion-exchange chromatography (see Fig. S4A to C) and measured concentration using a bicinchoninic acid (BCA) assay. The Sec:$SC^{(-)}$-$ELP_{60}$-336c strain secreted the largest amounts of protein, $60.3 \pm 2.22$ mg of $SC^{(-)}$-$ELP_{60}$-336c per liter of culture (Fig. 4C). Interestingly, the Sec:$SC^{(-)}$-$ELP_{60x}$-336c strain secreted roughly half as much as the $ELP_{60}$ variant ($31.6 \pm 1.07$ mg/liter culture, $P < 0.001$), and Sec:$SC^{(-)}$-$RLP_{12}$-336c secreted approximately one-sixth as much protein ($10.7 \pm 1.91$ mg/liter

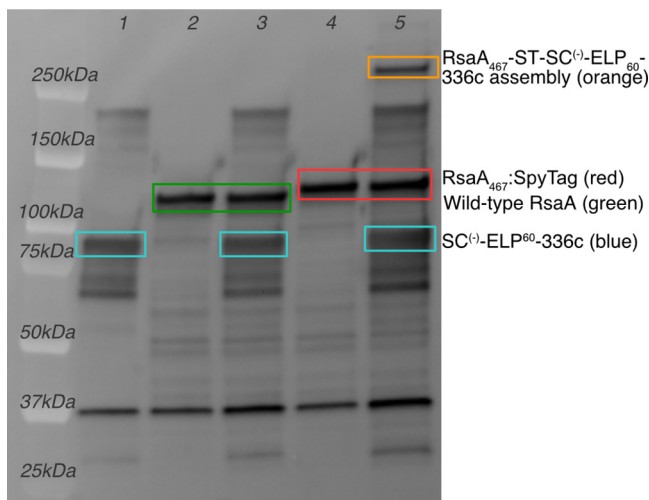

**FIG 5** Purified SC$^{(-)}$-ELP60-336c can be covalently bound to RsaA$_{467}$-ST on displayer cells. A high-molecular-weight band corresponding to the covalent assembly of purified SpyCatcher$^{(-)}$-ELP$_{60}$-336c protein and RsaA$_{467}$-SpyTag on engineered Disp:RsaA$_{467}$-ST cells is detected with anti-RsaA antibodies (lane 5, orange box). This band is not detected in control lanes containing only purified SC$^{(-)}$-ELP$_{60}$-336c protein (lane 1), only Disp:RsaA$_{wt}$ cells (lane 2), purified SC$^{(-)}$-ELP$_{60}$-336c protein with Disp:RsaA$_{wt}$ cells (lane 3), or only engineered Disp:RsaA$_{467}$-ST cells (lane 4).

culture, $P < 0.0001$). Significant differences in yield persist even when these values are normalized to the number of cells in the culture (Fig. S4D). Since ELP$_{60}$ variants and RLP$_{12}$ are predicted to have a disordered structure and suckerin$_{19}$ is predicted to be structured, these results support the hypothesis that secretion through T1SS is strongly affected by protein structure. Moreover, the high yields of secreted protein support *C. crescentus* as a multifunctional ELM chassis.

**Design and construction of a strain for covalent binding of extracellular matrix proteins.** We previously reported a *C. crescentus* CB15N strain capable of displaying SpyTag (ST) on the cell surface via insertion into the RsaA S-layer lattice (25). The variant with the highest level of SpyCatcher-mRFP1 surface binding, >11,000 copies per cell, contains SpyTag at amino acid location 467 of RsaA. For this study, we utilized the same SpyTag insertion but in the CB15 background as opposed to the CB15N background. CB15 is desirable for material formation as it retains the holdfast protein matrix at the end of the stalk structure, which allows *C. crescentus* stalked cells to adhere strongly to surfaces. Our rationale for using this background as opposed to CB15N is that this is an additional point of control contributing to the material properties, as it could allow for strong attachment of inorganic particles to our system. As before, we engineered a fluorescent protein into the CB15 background strain, this time, GFPmut3 (56), in place of SapA, which could undesirably edit our modified S-layer proteins. The final experimental construct is titled Disp:RsaA$_{467}$-ST, and a control strain that does not contain the SpyTag modification is titled Disp:RsaA$_{wt}$ (genotype details available in Fig. S1A). The SpyTag insertion was confirmed by colony PCR (Fig. S1B, lane 9).

**Extracellular matrix protein binds specifically and covalently to create a sEPM.** Next, we sought to test whether the synthetic extracellular matrix proteins were capable of ligation to SpyTag on the displayer strain, creating a sEPM. We incubated the Disp:RsaA$_{467}$-SpyTag with purified SC$^{(-)}$-ELP$_{60}$-336c protein and analyzed the reaction by immunoblotting with anti-RsaA polyclonal antibodies (Fig. 5). In control samples containing only SC$^{(-)}$-ELP$_{60}$-336c protein (Fig. 5, lane 1, 83-kDa band), Disp:RsaA$_{wt}$ alone (Fig. 5, lane 2, 112-kDa band), SC$^{(-)}$-ELP$_{60}$-336c protein with Disp:RsaA$_{wt}$ (Fig. 5, lane 3, 83-kDa and 113-kDa bands, respectively), or Disp:RsaA$_{467}$-ST cells alone (Fig. 5, lane 4, 121-kDa band), bands at 83 kDa or 112 kDa were detected, since all samples contain some RsaA, either the full RsaA protein (displayer cells) or the C-terminal 336c uncleaved secretion signal (purified protein). We note the apparent molecular weight

of the purified SC$^{(-)}$-ELP$_{60}$-336c protein is approximately 10 kDa lower than seen in secreted protein prior to purification. We attribute this change in gel migration pattern to charge screening of these proteins by the phosphate-buffered saline (PBS) solution, which may decrease aggregation and/or alter protein conformation (57, 58). A higher-molecular-weight band associated with covalent bonding between SC$^{(-)}$-ELP$_{60}$-336c and RsaA$_{467}$-ST in the control samples was not observed, which is consistent with the requirement for both SpyTag and SpyCatcher$^{(-)}$ to be present for ligation to occur. In the sample containing both SC$^{(-)}$-ELP$_{60}$-336c protein and Disp:RsaA$_{467}$-ST cells, a higher-molecular-weight product is apparent (Fig. 5, lane 5). To verify this assembly, we purified the S-layer protein from both Disp:RsaA$_{467}$-ST and Disp:RsaA$_{wt}$ cells and incubated the S-layer protein with and without purified SC$^{(-)}$-ELP$_{60}$-336c protein. Again, we only see the high-molecular-weight covalent assembly band in samples containing both SpyTag and SpyCatcher$^{(-)}$ moieties (see Fig. S6). These observations indicate that a specific covalent attachment is formed between the fusion protein and the Disp:RsaA$_{467}$-ST cell's engineered RsaA S-layer through the SpyCatcher$^{(-)}$-SpyTag system, resulting in hydrogel coating of the displayer strain and formation of a sEPM.

## DISCUSSION

As demonstrated above, we constructed a modular extracellular protein matrix through secretion of hydrogel materials that covalently coat cells. A switch to the supercharged SpyCatcher variant [SC$^{(-)}$] enables the extracellular matrix proteins to be secreted via a T1SS at unprecedented levels. Extracellular matrix proteins with hydrogel domains of an elastin-like polypeptide (ELP$_{60}$ and ELP$_{60x}$) or a resilin-like polypeptide (RLP$_{12}$) can be secreted, demonstrating the modularity of our approach. The extracellular matrix protein binds specifically to our engineered RsaA$_{467}$-SpyTag S-layer, enveloping the outermost cell surface and creating a sEPM. In the following, we discuss how our findings impact our understanding of type I secretion in *C. crescentus* and new routes toward self-coating bacteria and autonomous assembly of engineered living materials.

**Engineered *C. crescentus* is a platform for high-level secretion of biopolymers.** This work achieved unprecedented levels of biomaterial secretion by a Gram-negative type I secretion system and has the added benefit of being a genome-integrated system, which is more robust than plasmid-based systems (59). In our research, we discovered that through the switch of SpyCatcher to SpyCatcher$^{(-)}$, we achieve secretion of heterologous polymer-protein fusions (Fig. 3B), accomplishing the highest reported yields (60.3 ± 2.22 mg/liter) of a secreted biopolymer [SC$^{(-)}$-ELP$_{60}$-336c] by a Gram-negative bacterium (38) (Fig. 4B). We hypothesize that SpyCatcher$^{(-)}$ fusions are required in this system because SpyCatcher$^{(-)}$ remains largely disordered until it partners with SpyTag and the T1SS machinery has a strong preference for unfolded proteins. This hypothesis is further supported by the fact that we are unable to secrete fusions involving the suckerin$_{19}$ protein, as it contains structured beta sheets (51, 52, 60).

Moreover, our observations that the different polymer-protein fusions vary in secretion yields (Fig. 4C) uncovers design strategies for maximizing heterologous protein secretion through *C. crescentus*'s T1SS. For instance, SC$^{(-)}$-RLP$_{12}$-336c is secreted at significantly lower levels than SC$^{(-)}$-ELP$_{60}$-336c. Previous studies have shown that ABC transport systems, such as the RsaD-F$_{a,b}$ T1SS used herein, have higher secretion yields with proteins with isoelectric points (pIs) lower than 5.5. This pI selectivity is ascribed to the conformational changes of the transport machinery when it interacts with the target protein and the electric potential of cell membranes (61). While all of the successfully secreted proteins have an overall pI lower than 5.5, the lowest pI being 3.83 for 336c and the highest pI being 5.09 for SC$^{(-)}$-ELP$_{60x}$-336c, the pIs of the hydrogel domains within the full-length proteins vary greatly (see Table S3 in the supplemental material). Thus, we attribute the robust secretion yield of SC$^{(-)}$-ELP$_{60}$-336c to the ELP$_{60}$ domain's pI of 5.5 and the low secretion yield of SC$^{(-)}$-RLP$_{12}$-336c to the high pI of 9.91 for the RLP$_{12}$ domain. The ELP$_{60x}$ domain also has a high pI of 10.70, and, accordingly,

secretion levels of SC$^{(-)}$-ELP$_{60x}$-336c are lower than those of SC$^{(-)}$-ELP$_{60}$-336c (Fig. 4C). We also postulate that SpyCatcher was able to be secreted (Fig. 2B) despite that it is a folded protein because of its low pI of 4.14. This result confirms previous work showing that secretion of ELPs is affected by amino acid sequence, credited to the shift in surface chemistry interactions (62). Overall, our work corroborates previous efforts regarding high-yield secretion in the T1SS. Therefore, we suggest the following guidelines to achieve high-yield secretion. The target protein should have (i) minimal or, ideally, no regions with secondary or tertiary structure, (ii) an overall pI lower than 5.5, and (iii) individual domains with pIs lower than 5.5.

Our work indicates secreted biopolymers can easily be purified from *C. crescentus* cultures through anion-exchange chromatography of the extracellular media, without the need for cell lysis. Since *C. crescentus* secretes few extracellular proteins, there are fewer contaminants to remove from the target protein. These advantages are beneficial for applications where extremely pure hydrogel material is desired without expensive processing, establishing *C. crescentus* as a powerful chassis for the secretion of different biopolymers.

**sEPM-forming consortia have applications in biomanufacturing and engineered living materials.** We found that the SC$^{(-)}$-ELP$_{60}$-336c biopolymer binds covalently to the cell surface of our *C. crescentus* displayer strain via an engineered S-layer array. While there are many reports of cell encapsulation in chemically produced hydrogels (14, 16, 63–65), this is the first report of entirely bacterially synthesized covalent layering of hydrogel material on a bacterial cell surface. The use of spontaneously aggregating polymers such as ELP and RLP advances our previously reported two-dimensional (2D) assembly of a biomaterial onto a cell surface (25) into a 3D material. The potential for direct control over covalent and noncovalent interactions offered by this novel sEPM opens the door for the rational engineering of ELMs capable of self-encapsulation and autonomous assembly (7), i.e., not requiring intervention, induction chemicals, or applied force. Additionally, such a material could be formed by an oligotroph that does not require complex nutrients due to its high synthetic capacity (33, 66), making it a uniquely low-cost, low-effort living material. The functionality and physical properties of these ELMs would also be modular based on the biopolymer design and consortium dynamics. This platform could be applied to streamlined biomanufacturing of complex materials but also chemicals or fuels, as enveloping cells in a hydrogel can assist with cell protection in large-scale production. We recognize however that protein degradation/aggregation events may be a limiting factor for processing and downstream applications, as they may alter the percentage of biomaterial covalently bound or the structure of the hydrogel. Therefore, efforts spent optimizing conditions and identifying increasingly stable protein sequences would be beneficial before proceeding with large-scale material synthesis.

In addition to self-encapsulation for biomanufacturing, we suggest this approach has additional benefits for ELMs. First, by localizing the matrix protein to the cell surface, a hydrogel-like biomaterial may assemble under lower solution-phase protein concentrations than is usually reported for the creation of a strong hydrogel. Second, this displayer strain has been engineered in a background strain of *C. crescentus* that still retains the holdfast matrix at the base of its stalk as opposed to the secretor strain, in which this holdfast is no longer present. This allows for displayer cells to adhere strongly to surfaces and could be used to integrate inorganic materials into a hybrid material, such as for the introduction of orthogonal mechanical or optoelectronic properties (14, 67). The holdfast also opens the door to cell patterning through chemical modification of the surface (68) or layer-by-layer deposition of the ELM through bioprinting (69–71). This hierarchical assembly and cell patterning can lead to mechanical properties found in natural systems such as tolerance of compressive force (72) or arresting crack propagation (73, 74). Third, since the sEPM is self-synthesized, damage to the hydrogel layer can be continually repaired, the material can expand over time, and a small sample of the ELM can nucleate the growth of more material. Fourth, *C. crescentus* is nonpathogenic, has lower endotoxin activity than *E. coli* (75), and was

previously developed as a microbicide (27), making it a safe option for deployment. Fifth, while consortia have been used in only a few ELMs (76, 77), they allow for a division of labor between cell types, leading to more complex functionality and increased robustness (78, 79). A specific benefit of our two-strain system is that by starting with a single parent species, the cell growth rates are similar and there is no interspecies incompatibility. Upon further engineering control over the cell patterning within the consortia, we also envision usage for this advanced material in self-healing infrastructure, soft robotics, bioremediation, and biomedicine.

In summary, we describe the creation of strains that secrete a synthetic extracellular protein matrix and demonstrate the cell-surface attachment of the sEPM. In doing so, we confirmed guidelines for maximal biopolymer secretion via T1SS, including encoding an isoelectric point of ≤5.5 for each domain and the entire protein and limiting folded domains. Similar to a naturally occurring biofilm matrix, our engineered matrix is both composed of hydrogel-forming biomolecules and encodes specific binding to engineered strains. However, our engineered matrix binds through covalent bonds rather than weak interactions (3), and the matrix composition can potentially be altered to provide emergent properties. This work further develops *C. crescentus* as a chassis for high-level secretion, demonstrates a sEPM, and takes an important step forward toward creating autonomous ordered ELMs for use in biomanufacturing and advanced materials.

## MATERIALS AND METHODS

**Strains.** All strains used in this study are listed in Table S1 in the supplemental material. The *C. crescentus* strains were grown in PYE medium (0.2% peptone, 0.1% yeast extract, 1 mM $MgSO_4$, 0.5 mM $CaCl_2$) at 20°C or 30°C and with aeration at 250 rpm. The *E. coli* strains were grown in LB medium (1% tryptone, 0.5% yeast extract, 1% NaCl) at 37°C with aeration at 250 rpm. Depending on the strain, antibiotics were used at the following concentrations: for *E. coli*, 50 $\mu$g/ml ampicillin and 30 $\mu$g/ml kanamycin; for *C. crescentus*, 25 $\mu$g/ml kanamycin (plate). For conjugation methods, 300 $\mu$M diaminopimelic acid (DAP) was supplemented. For recombination methods, 3% (wt/vol) sucrose was supplemented. All chemicals were purchased from Sigma-Aldrich or VWR.

**Plasmid construction.** The list of all plasmids and primers used in the present study is available in Table S2. Details on construction of pNPTS138 integration plasmids can be found in Text S1. Plasmids were introduced to *E. coli* using standard transformation techniques with NEB 5-alpha chemically competent cells (New England BioLabs) and to *C. crescentus* using conjugation via *E. coli* strain WM3064.

**Genome engineering of *C. crescentus*.** To make the background strain (*C. crescentus* CB15N Δ*sapA*::Pxyl-*mkate2*), the *sapA* gene (CCNA_00783) was replaced with the gene for the mKate2 fluorescent protein under a xylose induction promoter. To achieve this, the 2-step recombination technique with sucrose counterselection method was employed. The fusion gene sequences for all synthetic extracellular matrix proteins and control proteins were integrated into the genome in place of the native *rsaA* gene using a 2-step recombination technique with sucrose counterselection, leaving the native regulatory sequence intact.

The 2-step recombination technique with sucrose counterselection is as follows: the pNPTS138 plasmids were electroporated into *E. coli* WM3064 cells and subsequently conjugated overnight into *C. crescentus* CB15N Δ*sapA*::Pxyl-*mkate2* on a PYE agar plate containing 300 $\mu$M DAP. The culture was then plated on PYE agar with kanamycin to select for integration of the plasmid and removal of WM3064 cells. Successful integrants were incubated in liquid PYE medium overnight and plated on PYE agar supplemented with 3% (wt/vol) sucrose to select for excision of the plasmid and *sacB* gene, leaving the target sequence in the genome. Integration of the sequences and removal of *sacB* gene were confirmed by colony PCR (Fig. S1B) with OneTaq Hot Start Quick-Load 2× master mix with GC buffer (New England BioLabs) using a Touchdown thermocycling protocol with an annealing temperature ranging from 72 to 62°C, decreasing 1°C per cycle. Primer sets for colony PCR verification can be found in Table S2.

Protein expression and secretion were evaluated in cultures of 25 ml of PYE medium with 0.02% Antifoam 204 (Sigma-Aldrich) after a 16-h incubation at 20°C with aeration at 250 rpm from a starting optical density at 600 nm ($OD_{600}$) of 0.02. Protein levels were evaluated through analysis of the whole-cell lysate and extracellular medium after separation by centrifugation at 10,000 relative centrifugal force (RCF) for 10 min. Fractions were combined with an equivalent volume of 4× Laemmli buffer (Bio-Rad), incubated for 20 min at 95°C, and then run on a Bio-Rad Criterion Stain-free 4% to 20% SDS-PAGE gel. The SDS-PAGE gel was transferred to a nitrocellulose membrane using the Bio-Rad Trans-Blot Turbo system and blocked with Thermo Fisher SuperBlock buffer for 1 h with agitation. The membrane was then washed several times with Tris-buffered saline with 0.1% Tween 20 (TBST) before a 30-min incubation with monoclonal mouse anti-FLAG M2 horseradish peroxidase (HRP)-conjugated antibodies (Sigma-Aldrich) diluted 1:5,000 in TBST. Thermo Fisher SuperSignal West Pico chemiluminescent substrate was used to activate HRP fluorescence, and the membrane was imaged under chemiluminescent mode in a ProteinSimple FluorChem E system. For SDS-PAGE gels, the Bio-Rad Precision Plus Protein unstained

protein standard was used. For immunoblot membranes, the Bio-Rad Precision Plus Protein Kaleidoscope prestained protein standard was used. Molecular-weight-band quantification was made using GelAnalyzer 19.1 software.

To determine if the extracellular protein was secreted or released due to cell lysis, we visualized the intracellular cell cycle transcriptional regulator (CtrA) in the whole-cell lysate and extracellular medium fractions. This was evaluated in cultures of 25 ml of PYE medium with 0.02% Antifoam 204 (Sigma-Aldrich) after a 16-h incubation at 20°C with aeration at 250 rpm from a starting $OD_{600}$ of 0.02. Protein levels were evaluated through analysis of the whole-cell lysate and extracellular medium after separation by centrifugation at 10,000 RCF for 10 min. Fractions were combined with equivalent volumes of 4× Laemmli buffer (Bio-Rad), incubated for 20 min at 95°C, and then run on a 12% Tris-glycine polyacrylamide gel (Bio-Rad 30% acrylamide/Bis solution). The gel was transferred to an Immun-Blot LF polyvinylidene difluoride (PVDF) membrane (Bio-Rad) and blocked with Thermo Fisher SuperBlock buffer for 1 h with agitation. The membrane was then washed several times with Tris-buffered saline with 0.1% Tween 20 (TBST) before an overnight incubation at 4°C with rabbit anti-CtrA antibodies (1:5,000) (80). After three 10-min TBST washes, the membrane was incubated with goat anti-rabbit HRP-conjugated secondary antibody (1:5,000) for 1 h. Western Lightning Plus-ECL (Perkin-Elmer) was used to activate HRP fluorescence, and the membrane was imaged under chemiluminescent mode in a Bio-Rad ChemiDoc MP. Spectra multicolor broad-range protein ladder (Thermo Fisher) was used.

**Protein quantification.** To quantify protein secretion, engineered *C. crescentus* strains were cultured for 24 h in PYE medium with 0.02% Antifoam 204 at 30°C with aeration at 250 rpm from a starting $OD_{600}$ of 0.02. 25 ml cultures were used for Sec:SC$^{(-)}$-RLP$_{12}$-336c, and 250 ml cultures were used for Sec:SC$^{(-)}$-ELP$_{60}$-336c and Sec:SC$^{(-)}$-ELP$_{60x}$-336c.

After incubation, the cultures were centrifuged for 20 min (8,000 RCF for 250 ml cultures, 5,250 RCF for 25 ml cultures) to extract the extracellular solution. The extracellular medium was then diluted 2-fold with 20 mM HEPES buffer at pH 7.0 and applied to 1 column volume (CV) of DEAE Sepharose fast flow resin (from GE Healthcare) equilibrated with 10 CV of 20 mM HEPES buffer at pH 7.0 (1 CV equaled 2 ml of settled resin for 25 ml cultures, and 1 CV equaled 4 ml of settled resin for 250 ml cultures). The supernatant was allowed to flow through the resin by gravity. The resin was subsequently washed with 10 CV of 20 mM HEPES buffer containing 50 mM NaCl at pH 7.0, and the protein was eluted with 3 CV of 20 mM HEPES buffer containing 500 mM NaCl at pH 7.0. This eluted fraction was placed in 12 to 14 kDa molecular weight cutoff regenerated cellulose dialysis tubing (Spectra/Por) and dialyzed in phosphate-buffered saline (PBS) overnight at 4°C with stirring.

Protein quantification was performed using the protocol and supplies provided by the BCA protein assay kit (Thermo Scientific, Pierce). Samples of BSA protein in PBS ranging in concentration from 25 $\mu$g/ml to 2,000 $\mu$g/ml were used to create the standard concentration curve. Triplicate measurements were obtained for each sample to ensure consistent concentration readings. The determined concentration value was then multiplied by the total sample volume to obtain the protein yield value for the culture, and this value was used to extrapolate the equivalent protein yield from a 1-liter culture. Statistical significance was determined by an unpaired, two-tailed Student's *t* test on Rstudios.

Optical quantification of protein bands was performed by densitometry using ImageJ software.

**Supercharged SpyCatcher-SpyTag binding assay.** Displayer strains containing either *rsaA*$_{wt}$ or *rsaA*$_{467}$:ST genes were grown at 30°C until they reached mid-log phase ($OD_{600}$ of 0.2 to 0.4). Cells ($10^8$) were harvested from each culture (as determined through optical density readings, where 1 ml of $OD_{600}$ of 0.5 is equivalent to $10^9$ cells) and resuspended in 100 $\mu$l of PBS and 0.5 mM $CaCl_2$. Purified SC$^{(-)}$-ELP$_{60}$-336c protein was then added to either Disp:RsaA$_{467}$-ST cells or Disp:RsaA$_{wt}$ cells at a ratio of 1 RsaA monomer (81) to 50 SC$^{(-)}$-ELP$_{60}$-336c monomers assuming 45,000 RsaA monomers per cell. In addition, three other negative controls were tested: Disp:RsaA$_{wt}$ cells without protein, Disp:RsaA$_{467}$-ST cells without protein, and SC$^{(-)}$-ELP$_{60}$-336c protein without cells. All five of these mixtures were incubated for 3 days at 4°C with agitation, as this temperature was previously shown to increase SpyCatcher$^{(-)}$ reactivity toward SpyTag (44). Afterwards, 15 $\mu$l from each sample was combined with an equivalent volume of 2× Laemmli buffer (Bio-Rad) and boiled at 98°C for 20 min. The samples were analyzed via immunoblot as described above except using polyclonal rabbit-anti-C-terminal RsaA antibodies (courtesy of the Smit lab, UBC [42]) diluted 1:5,000 in TBST followed by goat anti-rabbit IgG antibodies (HRP conjugate; Sigma-Aldrich) diluted 1:5,000 in TBST.

**S-layer extraction and SDS-PAGE analysis.** Disp:RsaA$_{467}$-ST-SC$^{(-)}$-ELP$_{60}$-336c cells with the appropriate negative controls (Disp:RsaA$_{467}$-ST, Disp:RsaA$_{wt}$, and Disp:RsaA$_{wt}$ with ST-SC$^{(-)}$-ELP$_{60}$-336c) were generated in triplicates according to the protocol for the supercharged SpyCatcher-SpyTag binding assay described above, scaled up to a starting count of $4.67 \times 10^8$ cells. Adapting the protocol developed by Walker et al. (82), cells were pelleted and washed twice with 750 $\mu$l of cold 10 mM HEPES buffer, pH 7.2. Cells were then pelleted, resuspended in 150 $\mu$l of cold 100 mM HEPES, pH 2, and incubated on ice for 5 min, which extracts the S-layer proteins from the cell surface. Following incubation, the solution was neutralized with 10 M NaOH, adding 3.5 $\mu$l of NaOH per 150 $\mu$l of 100 mM HEPES used; the supernatants of each sample were then transferred to Amicon Ultra 0.5-ml filters (MilliporeSigma) and concentrated to approximately 50 $\mu$l via centrifugation at 4°C.

From the concentrated solution, 15 $\mu$l was then mixed with an equal volume of 2× Laemmli buffer (Bio-Rad) and warmed to 40°C for 5 min before being analyzed by SDS-PAGE, using Criterion TGX precast gels (Bio-Rad). The Precision Plus Protein unstained protein standard (Bio-Rad) was used to quantify protein molecular weight. Gels were incubated for 5 min in double-distilled water (ddH$_2$O) three times under low-shaking conditions to remove SDS before incubation in GelCode blue stain reagent (Thermo Fisher Scientific) for 1 h under low-shaking conditions. Lastly, the protein stain was drained, and the gel

was incubated in ddH$_2$O for an additional hour to remove unbound stain. The gel was imaged using a ProteinSimple FluorChem M system.

## SUPPLEMENTAL MATERIAL

Supplemental material is available online only.

**TEXT S1**, DOCX file, 0.1 MB.
**FIG S1**, TIF file, 0.4 MB.
**FIG S2**, TIF file, 0.3 MB.
**FIG S3**, TIF file, 0.3 MB.
**FIG S4**, TIF file, 0.6 MB.
**FIG S5**, TIF file, 0.2 MB.
**FIG S6**, TIF file, 0.8 MB.
**TABLE S1**, DOCX file, 0.1 MB.
**TABLE S2**, DOCX file, 0.1 MB.
**TABLE S3**, DOCX file, 0.1 MB.

## ACKNOWLEDGMENTS

We thank John Smit and John Nomellini for helpful conversations and starting materials. We also thank Vera Troselj for technical assistance.

This work was supported by the Defense Advanced Research Projects Agency (Engineered Living Materials program to C.M.A.-F.). Work at the Molecular Foundry was supported by the Office of Science, Office of Basic Energy Sciences, of the U.S. Department of Energy under contract no. DE-AC02-05CH11231.

M.C., B.R., and C.M.A.-F. contributed to Conceptualization. M.C., M.T.O.-H., and N.T. contributed to Investigation, Methodology, and Visualization. D.L. and S.M. contributed to Investigation and Methodology. M.T.O.-H., R.F.T., D.L., and S.M. contributed Validation. K.R.R. and B.R. contributed Resources. M.C., K.R.R., P.D.A., C.M.A.-F., and B.R. contributed to Project Administration and Supervision. P.D.A. and C.M.A.-F. contributed to Funding Acquisition. M.C., M.T.O.-H., N.T. and C.M.A.-F. contributed to Writing – original draft. All authors contributed to Writing – review and editing.

We declare no conflicts of interest.

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
