## [Reviewer comments · mSystems]

Engineering high-yield biopolymer secretion creates an extracellular protein matrix for living materials

Maria Orozco-Hidalgo, Marimikel Charrier, Nicholas Tjahjono, Dong Li, Robert Tesoriero, Sara Molinari, Kathleen Ryan, Paul Ashby, Behzad Rad, and Caroline Ajo-Franklin

Corresponding Author(s): Caroline Ajo-Franklin, Rice University

Review Timeline:

Submission Date:	September 4, 2020
Editorial Decision:	October 7, 2020
Revision Received:	January 25, 2021
Accepted:	February 13, 2021

Editor: Danielle Tullman-Ercek

Reviewer(s): Disclosure of reviewer identity is with reference to reviewer comments included in decision letter(s). The following individuals involved in review of your submission have agreed to reveal their identity: Manuel Salmeron-Sanchez (Reviewer #2)

Transaction Report:

DOI: <https://doi.org/10.1128/mSystems.00903-20>

October 7, 2020

Dr. Caroline M. Ajo-Franklin
Rice University
BioSciences
6100 Main St.
MS 140
Houston, TX 77030

Re: mSystems00903-20 (Engineering high-yield biopolymer secretion creates an extracellular protein matrix for living materials)

Dear Dr. Ajo-Franklin:

Thank you for your recent manuscript submission to mSystems. Following careful review, I am happy to consider your manuscript for publication with minor modifications. Two peer reviewers made several suggestions for how to improve the manuscript, which you will find appended below. I look forward to receiving a revised version of the manuscript that addresses their comments at your earliest convenience; we are flexible with timing due to the constraints of the pandemic but do let me know if you expect revisions to take longer than 6 weeks. In addition, when submitting your revisions, please reduce the number of supplemental files provided to less than 10. Reach out to me if you have any questions on how to do so.

To submit your modified manuscript, log onto the eJP submission site at <https://msystems.msubmit.net/cgi-bin/main.plex>. If you cannot remember your password, click the "Can't remember your password?" link and follow the instructions on the screen. Go to Author Tasks and click the appropriate manuscript title to begin the resubmission process. The information that you entered when you first submitted the paper will be displayed. Please update the information as necessary. Provide (1) point-by-point responses to the issues raised by the reviewers as file type "Response to Reviewers," not in your cover letter, and (2) a PDF file that indicates the changes from the original submission (by highlighting or underlining the changes) as file type "Marked Up Manuscript - For Review Only."

Due to the SARS-CoV-2 pandemic, our typical 60 day deadline for revisions will not be applied. I hope that you will be able to submit a revised manuscript soon, but want to reassure you that the journal will be flexible in terms of timing, particularly if experimental revisions are needed. When you are ready to resubmit, please know that our staff and Editors are working remotely and handling submissions without delay. If you do not wish to modify the manuscript and prefer to submit it to another journal, please notify me of your decision immediately so that the manuscript may be formally withdrawn from consideration by mSystems.

Corresponding authors may join or renew ASM membership to obtain discounts on publication fees.

Need to upgrade your membership level? Please contact Customer Service at Service@asmusa.org.

Thank you again for submitting your paper to mSystems.

Sincerely,

Danielle Tullman-Ercek

Editor, mSystems

Journals Department
Reviewer comments:

Reviewer #1 (Comments for the Author):

Review for

"Engineering high-yield biopolymer secretion creates an extracellular protein matrix for living materials"

Charrier et al.

In this study, the authors engineer a two-strain *Caulobacter* system in which one strain secretes an elastin-like (or resilin-like) peptide fused to a Supercharged SpyCatcher (SC-ELP), and the other strain displays the SpyTag on its surface, which can covalently bind the secreted SC-ELP peptide. They demonstrate a high level of heterologous protein secretion (60 mg/L) by the *Caulobacter* type I secretion system with a neat application: growth of a programmable extracellular matrix on a specific cell strain in culture. They also reveal some biophysical factors that influence heterologous type I secretion.

Overall, this is a nice study. The authors show convincing evidence that they can secrete ELPs or RLPs and that these bind to their "displayer" strain. However, it would be better to see more conclusive evidence for their statement that the pI of the protein to be secreted determines secretion efficiency, and to provide more information about some of the results with a few follow-up experiments, as follows.

Comments:

1) The authors repeatedly claim an unprecedented level of biopolymer secretion from a Gram-negative bacterium at 60 mg/ml. This is false - see AA Glasgow, HT Wong, and D Tullman-Ercek, ACS Synthetic Biology 2017. This study shows type 3 secretion in *Salmonella* of biopolymer proteins at 2X the titer reported here, by strain engineering and the addition of an exogenous protein to culture. The authors should therefore tone down this claim. I don't think this takes away

from the impact of this work, though. Here, the authors engineered a system that uses a different secretion system in a quite different organism, and they also reached high titers. It is also impressive that their system is fully genome-integrated, which is more robust than a plasmid-based expression system.

2) The blots show substantial degradation products (Fig. 4B) and aggregation products (if this is what the high band in Fig. S5 is - not sure because band is not annotated in the main text or the figure caption). Were these included in the reported values for secretion titers, as measured by BCA? If so, please correct.

3) Given that expression of biopolymers is prone to degradation/aggregation in this system (Fig. 2B), can the authors please add a few lines to the discussion to comment on how that impacts downstream uses for the proteins in this system and others?

4) Relatedly, it would be useful if the authors quantified:

- how much by percentage of the expressed protein is secreted in each strain
- how much of the total protein content of the supernatant is the ELP or RLP, since *Caulobacter* is secreting native proteins as well
- how much of the ELP/RLP in the supernatant is definitely from secretion and not from cell lysis by blotting against an intracellular protein (in my opinion this is an important control)
- if possible, what fraction of secreted full-length SC-ELP binds to the "displayer" strain to yield sEPM.

5) Finally, it is very interesting that the less-structured, more negatively charged Supercharged SpyCatcher can be secreted with ELP60, but not regular SpyCatcher, and that there is some effect from protein pI on secretion titer also. However, in order to make these strong statements about specific design criteria for secreting proteins by this system in the discussion section, the authors should either include a graph showing a correlation between pI and secretion titer for the proteins that they already secreted, or test the effects of low pI and/or the inclusion of structured domains for these heterologous biopolymer proteins more systematically in an additional experiment. For example, does adding an aspartic acid tail to Suckerin render it secretable? Would this approach improve the titer of RLP secretion to that of the ELP, as per H Byun et al., *J. Biol. Chem.* 2017? I think that if the authors don't want to do an experiment like this, that's OK, but they should then simply say that their results agree with previous work about factors that influence type I heterologous secretion and not frame this agreement as design criteria, which should be more specific to be useful.

Thank you for the opportunity to read about this cool research in a clearly written paper!

Reviewer #2 (Comments for the Author):

The paper by Charrier et al shows a system to engineer ECMs using biopolymers and the SpyCatcher/SpyTag technology. The field of living materials is important and is gaining traction as potential new materials for a growing number of applications. The paper is well written and easy to follow and the experiments that are included are relevant and well performed.

Comments:

The fact that authors put emphasis on the engineering of living materials, raise some questions

about the practical applications of the system. This is a complex bacterial system that includes two genetically engineered strains. It'd be beneficial to put some more emphasis on the impact that the work might have in the community of engineered living materials and beyond.

The covalent attachment between the secreted 'hydrogel molecules' and the engineered 'displayer' is not strongly demonstrated. The covalent linkage is important and authors are encouraged to demonstrate this beyond the observations they show in Figure 4

The ECM is a crosslinked hydrogel of interconnected fibrils. I wonder how a secreted hydrogel which contains a terminal SpyCather group - and then is only able to react through one side of the molecule - can form an interconnected ECM network. If covalent attachment happens between the secreted SpyCather hydrogel and the engineered SpyTag strain, one can imagine the displayer coated with a layer of 'hydrogel molecules'. Why should this lead to an interconnected ECM?

To demonstrate the importance of the SpyCather group in the formation of ECM, it'd be interesting to perform nanomechanical characterisation (e.g. Atomic Force Microscopy or similar) of the system to assess (1) the mechanical properties of the engineered secreted materials, (2) the role of the covalent binding of the secreted hydrogel molecules (w/o SpyCather group, different hydrogel molecules, molecular weight, etc)

Review for

“Engineering high-yield biopolymer secretion creates an extracellular protein matrix for living materials”
Charrier et al.

In this study, the authors engineer a two-strain *Caulobacter* system in which one strain secretes an elastin-like (or resilin-like) peptide fused to a Supercharged SpyCatcher (SC-ELP), and the other strain displays the SpyTag on its surface, which can covalently bind the secreted SC-ELP peptide. They demonstrate a high level of heterologous protein secretion (60 mg/L) by the *Caulobacter* type I secretion system with a neat application: growth of a programmable extracellular matrix on a specific cell strain in culture. They also reveal some biophysical factors that influence heterologous type I secretion.

Overall, this is a nice study. The authors show convincing evidence that they can secrete ELPs or RLPs and that these bind to their “displayer” strain. However, it would be better to see more conclusive evidence for their statement that the pI of the protein to be secreted determines secretion efficiency, and to provide more information about some of the results with a few follow-up experiments, as follows.

Comments:

- 1) The authors repeatedly claim an unprecedented level of biopolymer secretion from a Gram-negative bacterium at 60 mg/ml. This is false – see AA Glasgow, HT Wong, and D Tullman-Ereck, *ACS Synthetic Biology* 2017. This study shows type 3 secretion in *Salmonella* of biopolymer proteins at 2X the titer reported here, by strain engineering and the addition of an exogenous protein to culture. The authors should therefore tone down this claim. I don’t think this takes away from the impact of this work, though. Here, the authors engineered a system that uses a different secretion system in a quite different organism, and they also reached high titers. It is also impressive that their system is fully genome-integrated, which is more robust than a plasmid-based expression system.
- 2) The blots show substantial degradation products (Fig. 4B) and aggregation products (if this is what the high band in Fig. S5 is – not sure because band is not annotated in the main text or the figure caption). Were these included in the reported values for secretion titers, as measured by BCA? If so, please correct.
- 3) Given that expression of biopolymers is prone to degradation/aggregation in this system (Fig. 2B), can the authors please add a few lines to the discussion to comment on how that impacts downstream uses for the proteins in this system and others?
- 4) Relatedly, it would be useful if the authors quantified:
 - how much by percentage of the expressed protein is secreted in each strain
 - how much of the total protein content of the supernatant is the ELP or RLP, since *Caulobacter* is secreting native proteins as well
 - how much of the ELP/RLP in the supernatant is definitely from secretion and not from cell lysis by blotting against an intracellular protein (in my opinion this is an important control)
 - if possible, what fraction of secreted full-length SC-ELP binds to the “displayer” strain to yield sEPM.
- 5) Finally, it is very interesting that the less-structured, more negatively charged Supercharged SpyCatcher can be secreted with ELP60, but not regular SpyCatcher, and that there is some effect from protein pI on secretion titer also. However, in order to make these strong statements about specific design criteria for secreting proteins by this system in the discussion section, the authors should either include a graph showing a correlation between pI and secretion titer for the proteins

that they already secreted, or test the effects of low pI and/or the inclusion of structured domains for these heterologous biopolymer proteins more systematically in an additional experiment. For example, does adding an aspartic acid tail to Suckerin render it secretable? Would this approach improve the titer of RLP secretion to that of the ELP, as per H Byun *et al.*, *J. Biol. Chem.* 2017? I think that if the authors don't want to do an experiment like this, that's OK, but they should then simply say that their results agree with previous work about factors that influence type I heterologous secretion and not frame this agreement as design criteria, which should be more specific to be useful.

Thank you for the opportunity to read about this cool research in a clearly written paper!

Response to Reviewers for "Engineering high-yield biopolymer secretion creates an extracellular protein matrix for living materials"

Reviewer #1

In this study, the authors engineer a two-strain *Caulobacter* system in which one strain secretes an elastin-like (or resilin-like) peptide fused to a Supercharged SpyCatcher (SC-ELP), and the other strain displays the SpyTag on its surface, which can covalently bind the secreted SC-ELP peptide. They demonstrate a high level of heterologous protein secretion (60 mg/L) by the *Caulobacter* type I secretion system with a neat application: growth of a programmable extracellular matrix on a specific cell strain in culture. They also reveal some biophysical factors that influence heterologous type I secretion.

Overall, this is a nice study. The authors show convincing evidence that they can secrete ELPs or RLPs and that these bind to their "displayer" strain. However, it would be better to see more conclusive evidence for their statement that the pI of the protein to be secreted determines secretion efficiency, and to provide more information about some of the results with a few follow-up experiments, as follows.

Response: We thank the Reviewer for the Reviewer's endorsement of our work. To more accurately reflect the limitations of our experimental evidence, we have rephrased our assertions regarding the pI (lines 126, 379-380, 434-435).

Comments:

1) The authors repeatedly claim an unprecedented level of biopolymer secretion from a Gram-negative bacterium at 60 mg/ml. This is false - see AA Glasgow, HT Wong, and D Tullman-Ercek, ACS Synthetic Biology 2017. This study shows type 3 secretion in *Salmonella* of biopolymer proteins at 2X the titer reported here, by strain engineering and the addition of an exogenous protein to culture. The authors should therefore tone down this claim. I don't think this takes away from the impact of this work, though. Here, the authors engineered a system that uses a different secretion system in a quite different organism, and they also reached high titers. It is also impressive that their system is fully genome-integrated, which is more robust than a plasmid-based expression system.

Response: We thank the reviewer for bringing this work to our attention. We have modified our claim about the secretion yield to highlight that it was achieved via a Type I secretion system (lines 37-39, 103-105, 122-124, 353-355).

2) The blots show substantial degradation products (Fig. 4B) and aggregation products (if this is what the high band in Fig. S5 is - not sure because band is not annotated in the main text or the figure caption). Were these included in the reported values for secretion titers, as measured by BCA? If so, please correct.

Response: We have clarified that the reported secretion titers include degradation, aggregation and full length products (lines 276-285). Also, we have annotated the high MW band on Fig. S5 in the figure description section.

3) Given that expression of biopolymers is prone to degradation/aggregation in this system (Fig. 2B), can the authors please add a few lines to the discussion to comment on how that impacts downstream uses for the proteins in this system and others?

Response: We have included a paragraph addressing these concerns (lines 276-285, 406-410).

4) Relatedly, it would be useful if the authors quantified:
- how much by percentage of the expressed protein is secreted in each strain

Response: We have now included percentages of intracellular and extracellular protein in the SDS-PAGE samples shown in Fig. 2B and Fig. 4B using densitometry. We note that the extracellular sample only represents 0.006% of the total media volume, while the pellet sample represents 0.2% of the total cell volume. Because the protein has a tendency to self-aggregate, we cannot confidently state that the extracellular sample is representative of the much larger culture. Thus, we limit ourselves to the qualitative assessment that the majority of the protein is in the extracellular media. We have edited the text to include this data and briefly discuss it (lines 187-192, 257-261, 536-537).

- how much of the total protein content of the supernatant is the ELP or RLP, since *Caulobacter* is secreting native proteins as well

Response: The secreted biopolymers are the predominant protein in the supernatant. We have added a SDS-PAGE of the extracellular protein content of the Sec:SC(-)-ELP60-336c strain (Fig. S8) that shows the ELP biopolymer is the only protein visible. We have edited the main text to include this data (lines 265-269).

- how much of the ELP/RLP in the supernatant is definitely from secretion and not from cell lysis by blotting against an intracellular protein (in my opinion this is an important control)

Response: We thank the Reviewer for raising this important point. We have now included a cell lysis control by blotting against CtrA, a cell cycle transcriptional regulator located exclusively inside the cell envelope (Fig. S7). There is no measurable CtrA in the extracellular supernatant, indicating EPM is not released into the extracellular space because of cell lysis. We also edited the main text to describe these findings (lines 269-273, 496-511).

- if possible, what fraction of secreted full-length SC-ELP binds to the "displayer" strain to yield sEPM.

Response: We have attempted to quantify this interaction multiple ways, however, none of these approaches yielded consistent data on the fraction of SC-ELP bound. We tried to remove unbound ELP from Displayer-attached ELP via centrifugation or dialysis. We also attempted to image bound and unbound ELP via microscopy. The problem underlying all these failed efforts is that SC-ELP has a strong tendency to self-aggregate, especially at high concentration, thus limiting our ability to remove non-covalently bound SC-ELP. Only the harsh conditions of SDS-PAGE or S-layer extraction, which are incompatible with maintaining cell integrity, allowed us to remove non-covalently bound SC-ELP consistently (Figure 5 and Figure S9).

5) Finally, it is very interesting that the less-structured, more negatively charged Supercharged SpyCatcher can be secreted with ELP60, but not regular SpyCatcher, and that there is some effect from protein pI on secretion titer also. However, in order to make these strong statements about specific design criteria for secreting proteins by this system in the discussion section, the authors should either include a graph showing a correlation between pI and secretion titer for the proteins that they already secreted, or test the effects of low pI and/or the inclusion of structured domains for these heterologous biopolymer proteins more systematically in an additional experiment. For example, does adding an aspartic acid tail to Suckerin render it secretable? Would this approach improve the titer of RLP secretion to that of the ELP, as per H Byun et al., J. Biol. Chem. 2017? I think that if the authors don't want to do an experiment like this, that's OK, but they should then simply say that their results agree with previous work about factors that influence type I heterologous secretion and not frame this agreement as design criteria, which should be more specific to be useful.

Response: The Reviewer raises a fair point. We recognize that our claim regarding the role of pI in high titers of secreted protein is very strong, given the limited experimental evidence we provided. Henceforth, we have rephrased our conclusion, clarifying that our claim is a confirmation of previous work (lines 126, 379-380, 434-435).

Thank you for the opportunity to read about this cool research in a clearly written paper!

Reviewer #2

The paper by Charrier et al shows a system to engineer ECMs using biopolymers and the SpyCatcher/SpyTag technology. The field of living materials is important and is gaining traction as potential new materials for a growing number of applications. The paper is well written and easy to follow and the experiments that are included are relevant and well performed.

Response: We thank the Reviewer for the Reviewer's strong support of our work.

Comments:

The fact that authors put emphasis on the engineering of living materials, raise some questions about the practical applications of the system. This is a complex bacterial system that includes two genetically engineered strains. It'd be beneficial to put some more emphasis on the impact that the work might have in the community of engineered living materials and beyond.

Response: We thank the Reviewer for raising this important point. We have edited the main text to emphasize on the impact of our platform in ELMs (lines 297-406, 426-432).

The covalent attachment between the secreted 'hydrogel molecules' and the engineered 'displayer' is not strongly demonstrated. The covalent linkage is important and authors are encouraged to demonstrate this beyond the observations they show in Figure 4.

Response: We probed for covalent bond formation between the secreted hydrogel molecules and displayer cells using SDS-PAGE followed by immunoblotting. While SpyTag and SpyCatcher can bind to each other non-covalently, these interactions are disrupted by strongly denaturing conditions of SDS-PAGE, i.e. addition of 2% detergent and boiling for 15 m. Thus, SDS-PAGE shows a SpyTag-SpyCatcher conjugation product only when an isopeptide bond forms between SpyTag and SpyCatcher (Zakeri, B. et al. Proc. Natl. Acad. Sci. U. S. A. 109, E690–7 (2012)). Because of this, SDS-PAGE is the primary way covalent bond formation is probed in the Tag/Catcher literature, e.g. Zhang, W.-B., Sun, F., Tirrell, D. A. & Arnold, F. H. J. Am. Chem. Soc. 135, 13988–13997 (2013); Veggiani, G. et al. Proc. Natl. Acad. Sci. U. S. A. 113, 1202–1207 (2016).

Nonetheless, we performed new experiments to further corroborate this linkage. We purified the S-layer from the Disp:RsaA467-ST strain after its incubation with SC(-)-ELP60-336c purified protein. We analyzed these samples by SDS-PAGE, confirming the presence of a higher molecular weight product (Figure S9, lines 332-336, 554-573). Additionally, the S-layer purification protocol requires additional exposure to highly acidic conditions (pH 2). The fact that after subjection to these denaturing conditions we detected the higher molecular weight product, further supports the presence of an isopeptide bond.

The ECM is a crosslinked hydrogel of interconnected fibrils. I wonder how a secreted hydrogel which contains a terminal SpyCather group - and then is only able to react through one side of the molecule - can form an interconnected ECM network. If covalent attachment happens between the secreted SpyCather hydrogel and the engineered SpyTag strain, one can imagine the displayer coated with a layer of 'hydrogel molecules'. Why should this lead to an interconnected ECM?

Response: The Reviewer raises an important question: Does our engineered system have the potential to form an interconnected ECM network?

While only one terminus of the biopolymer is able to react with SpyTag on the cell surface, both additional non-covalent and covalent interactions can contribute to form an

interconnected matrix. We expect non-covalent hydrophobic interactions between the hydrogel proteins can lead to the formation of an interconnected matrix at higher protein concentrations. (Indeed, in work currently in preparation, we have observed hydrophobic interactions leading to an interconnected ECM network.) Furthermore, we have engineered a crosslinkable variant of the Sec: SC⁽⁻⁾-ELP₆₀-336c strain. This strain includes residues of glutamine and lysine that allow for enzymatic crosslinking via transglutaminase. We envision that adding the enzymatic crosslinker to the co-culture will increase the cohesiveness of the ECM network by creating new connections between biopolymers attached to different displayer cells.

We have added a few lines in the main text explaining how the interconnected ECM network can be formed (lines 138-141) and explaining the role of this strain (lines 143-145, 237-240).

To demonstrate the importance of the SpyCather group in the formation of ECM, it'd be interesting to perform nanomechanical characterisation (e.g. Atomic Force Microscopy or similar) of the system to assess (1) the mechanical properties of the engineered secreted materials, (2) the role of the covalent binding of the secreted hydrogel molecules (w/o SpyCather group, different hydrogel molecules, molecular weight, etc)

Response: The Reviewer raises a relevant point by suggesting nanomechanical characterization, nevertheless we consider this to be out of the scope of this paper. While we recognize the importance of these measurements in the ELMs community, this paper is focused on the design of ELMs platforms from a synthetic biology standpoint.

February 13, 2021

Dr. Caroline M. Ajo-Franklin
Rice University
BioSciences
6100 Main St.
Mailstop 140
Houston, TX 77005

Re: mSystems00903-20R1 (Engineering high-yield biopolymer secretion creates an extracellular protein matrix for living materials)

Dear Dr. Caroline M. Ajo-Franklin:

I am pleased to inform you that your manuscript has been accepted, and I am forwarding it to the ASM Journals Department for publication. For your reference, ASM Journals' address is given below. Before it can be scheduled for publication, your manuscript will be checked by the mSystems senior production editor, Ellie Ghatineh, to make sure that all elements meet the technical requirements for publication. She will contact you if anything needs to be revised before copyediting and production can begin. Otherwise, you will be notified when your proofs are ready to be viewed.

- Minimum resolution of 1280 x 720
- .mov or .mp4. video format
- Provide video in the highest quality possible, but do not exceed 1080p
- Provide a still/profile picture that is 640 (w) x 720 (h) max

We recognize that the video files can become quite large, and so to avoid quality loss ASM

suggests sending the video file via <https://www.wetransfer.com/>. When you have a final version of the video and the still ready to share, please send it to Ellie Ghatineh at eghatineh@asmusa.org.

Sincerely,

Danielle Tullman-Ercek
Editor, mSystems

Journals Department
Fig. S4: Accept

Fig. S2: Accept

Table S3: Accept

Fig. S1: Accept

Supplemental Methods: Accept

Fig. S6: Accept

Table S2: Accept

Table S1: Accept

Fig. S5: Accept

Fig. S3: Accept